# Associations between Self-Objectification and Lifestyle Habits in a Large Sample of Adolescents

**DOI:** 10.3390/children9071022

**Published:** 2022-07-08

**Authors:** Migle Baceviciene, Rasa Jankauskiene, Laima Trinkuniene

**Affiliations:** 1Department of Physical and Social Education, Lithuanian Sports University, 44221 Kaunas, Lithuania; laima.trinkuniene@lsu.lt; 2Institute of Sport Science and Innovations, Lithuanian Sports University, 44221 Kaunas, Lithuania; rasa.jankauskiene@lsu.lt

**Keywords:** self-objectification, eating attitudes and behaviours, dietary habits, smoking, alcohol use, physical activity, sedentary lifestyle, adolescents

## Abstract

The aim of the present study was to test associations between self-objectification and health-related lifestyle habits in a large sample of adolescents. In 2019–2020, a large sample of adolescents (*n* = 1402, 40.4% boys) participated in the survey and, as a part of a larger study, filled in questionnaires on self-objectification and lifestyle. Self-objectification was assessed using the Self-Objectification Questionnaire (SOQ). The lifestyle questionnaire had three batteries: eating attitudes and behaviors, dietary habits, and questions on harmful behaviors together with time spending patterns: sedentary lifestyle and physical activity. The ages ranged from 15–19 years with a mean age of 16.97 ± 0.46 years. Self-objectification (SO) was associated with a less healthy lifestyle: more frequent dieting and unhealthy eating habits in girls, skipping breakfast, and a lower number of meals per day. Adolescent boys and girls with higher SO demonstrated more frequent smoking and alcohol consumption until they felt dizzy and more frequent consumption of soft drinks, energy drinks, and fast foods. Girls with higher SO demonstrated lower perceived physical activity and longer duration of internet browsing for non-educational purposes. The results of the present study inform healthy lifestyle promotion programs for adolescents, suggesting that including psychoeducation about self-objectification, body functionality, and positive embodiment might be an effective strategy.

## 1. Introduction

According to the objectification theory [1], women and girls regularly experience being treated and evaluated as objects to be looked at. Therefore, their socialization is based on internalization of the observer’s perspective of their own bodies. This process is known as “self-objectification” [1,2]. Regular monitoring of the appearance and body surveillance (monitoring of the appearance from the outsiders’ perspective) promote body shame, which is related to the attitudes and feelings that the body does not conform to the sociocultural appearance standards. Body shame, body surveillance, and beliefs that one can control one’s own appearance comprise objectified body consciousness [2,3]. Self-objectification is described as valuing body appearance over body competence [4]. As opposite, appreciating body functionality and respecting the body for what it is capable of doing are important aspects of a positive body image [5,6]. Importantly, being in high trait self-objectification means diverting much attention to one’s appearance but not necessarily feeling dissatisfied with it [7]. The majority of studies concluded that women and girls experience higher self-objectification [2] and have a less positive body image compared to men and boys [8].

In adults, the self-objectification has been linked to a negative body image (i.e., body shame), disordered eating, higher use of social networking, internalization of stereotyped body ideals, social physique anxiety, lower self-esteem, depression, and appearance—related reasons to exercise [1,9,10,11,12,13,14,15,16,17,18,19,20]. The self-objectification theory is less tested on adolescents; nevertheless, existing studies suggested that self-objectification is associated with disordered eating, comments about body image, peer teasing, media consumption, internalization of appearance ideals, depressive symptoms, somatic complaints, and lower self-esteem [2,18,20,21,22,23]. A recent review concluded that more research on the associations between self-objectification and physical health is needed [2]. Finally, to the best of our knowledge, there is no research that has explored the associations between self-objectification and health-related lifestyle habits (eating habits, physical activity, smoking, and alcohol consumption) in adolescents. Based on the conceptions of a positive body image and body functionality, if a person appreciates their body, they love and care about their body, appreciate its functionality in a broad meaning, resist sociocultural pressures to achieve stereotyped beauty appearances, and have body flexibility, meaning that all types of bodies must be appreciated and loved. Therefore, they should be less likely to harm their body through unhealthy behaviours such as smoking, drinking alcohol, unhealthy eating, and living a sedentary lifestyle [5,6]. In contrast, valuing body appearance over body competence might be associated with more unhealthy lifestyle habits. However, there is a lack of studies testing this assumption in adolescents. The results of the present study might inform healthy lifestyle promoting programs for adolescents.

The aim of the present study was to test associations between self-objectification and health–related lifestyle habits in a large sample of adolescent girls and boys. In the present study, we expected that self-objectification or valuing appearance over body competence would be positively associated with unhealthy eating and dieting, smoking, alcohol consumption and negatively associated with leisure–time exercise, perceived physical fitness, and self-rated health. We also expected the correlations would be higher in adolescent girls than in boys.

## 2. Materials and Methods

### 2.1. Study Organization and Study Participants

Forty-one schools from 26 Lithuanian cities and towns participated in the study representing the main municipalities of the country. Study participants included 1412 schoolchildren of the 11th year: 842 were girls (59.6%) and 570 were boys (40.4%). The mean age was 16.94 ± 0.47 years (range of 15–19 years) and 17.02 ± 0.45 years (range of 15–19 years) for the girls and boys, respectively, and 82.4% were seventeen.

The study was conducted in November 2019 to January 2020 before the COVID-19 pandemic. The study was approved by the Social Research Ethics Committee of Lithuanian Sports University (Protocol No. SMTEK-32, 27 September 2019). After obtaining permission from the school principals, the anonymous online survey link was spread with the help of the schoolteachers. The survey was implemented after the lessons were over or during the breaks. Additionally, only schoolchildren whose parents or caregivers provided a verbal consent to participate in the survey were involved. After an introduction to the study aims, data confidentiality, and average duration of the survey, study participants were provided a digital consent with the options “I agree to participate” or “I disagree to participate”. Those who agreed were provided with the study measures. In cases where a disagreement was provided, study participants were acknowledged, and the survey was terminated. In addition, there was a possibility to stop the survey at any point by closing a browser without recording the answers.

The survey was implemented by using the Google Forms platform with an average duration of 25–30 min containing ≈130 items. The response rate was close to 90%. The online survey form was restricted to accept only one answer from the same IP address. There were 56 refusals to participate without the request to indicate the reason. Furthermore, 24 questionnaires were not used, as they were filled in incorrectly (for example, the indicated actual or desired body weight and height were not realistic). The final sample approved for statistical analysis contained 1412 subjects with no missing data, as all questions were set as mandatory.

### 2.2. Study Measures

The sociodemographic part of the questionnaire contained questions on city, school, gender, and age. Additionally, study participants were asked to provide their weight (kg) and height (cm) for the body mass index (BMI) calculation. The mean BMI was 20.96 ± 3.06 kg/m^2^ (range of 14–41.7 kg/m^2^) and 21.99 ± 2.99 kg/m^2^ (range of 15.4–40.6 kg/m^2^) in girls and boys, respectively. According to the International Obesity Task Force (IOTF) BMI cutoffs [24], 79.1% and 72.9% of boys and girls, respectively, were classified as normal weight.

The Self-Objectification Questionnaire (SOQ) [20,25] was used to assess the extent to which an individual sees his/her body in an objectified (appearance related) manner and/or in a non-objectified (body functionality related) manner. Study participants were provided ten different body features with five of them representing body appearance (sexuality, physical attractiveness, body measurements, body weight, and trained muscles) and the other five representing body competence (physical fitness, coordination, health, strength, and energy). Study participants were asked to score each body feature in order of importance from 0 (not important) up to 9 (extremely important), and the scores were summarized for Appearance and Competence subscales. The same score could be given only once. Thus, each body feature had to be attributed a different score. Finally, the scores for body competence attributes were subtracted from the appearance-based attribute scores, resulting in final scores ranging from −25 to +25, with higher scores indicating a greater emphasis on appearance and a higher trait of self-objectification. A positive score indicates that the person values their own body’s appearance over its functionality, whereas a negative score indicates that the person values functionality over appearance. Because all body features were scored at different values, it is not possible to calculate the Cronbach’s α. However, the construct validity of the Lithuanian version of the instrument has previously been demonstrated by positive correlations of the body competence attribute subscale with a more positive body image and lower disordered eating [26].

Study participants were asked to indicate their weighing frequency during the past month by providing the response options from “never” (0) up to “several times a day” (6) [27].

A single statement “I have tried to lose weight by fasting or going on crash diets” assessed dieting with the response options from “never” (0) up to “very often” (4) from the Multidimensional Body Self-Relations Questionnaire, Overweight Preoccupation subscale [28,29].

The unhealthy nutrition score was calculated from a 5-item questionnaire [30] asking about the frequency of eating in a rush, eating when reading or watching TV, eating late at night less than 2 h before sleep, overeating, and using unhealthy foods for snacking (candies, cakes, chips). The response options were from “never” (0) up to “always” (4). The final score was calculated by averaging the response options, with the higher value representing more frequent unhealthy eating habits.

Information on dietary habits was collected in a frequency style asking to indicate the frequency of each group consumption from “never” (0) up to “every day” (3). The original food frequency questionnaire used for this study and approved for Health Behaviour among Lithuanian Adult Population contained 20 food groups [30]. For the analysis, we used the representatives of healthy (fruits, berries, vegetables, porridge and cereals, fish) and unhealthy food groups (soft drinks, energy drinks, sweets and cakes, fast food).

Questions on smoking (“How often do you smoke tobacco at present?”) with the response options from “I do not smoke” (0) up to “every day” (3) and alcohol use (“Have you ever had so much alcohol in your life that you were really drunk?”) with the response options from “never” (0) up to “more than 10 times” (4) were taken from the nationally approved survey for Health Behavior in School-Aged Children (HBSC) [31].

Sleep duration in hours per usual workday was indicated by the study participants [32].

The duration of internet browsing for non-educational purposes was assessed by a single question asking to indicate the mean time in hours per usual workday, which was used in our previous studies [33].

Perceived physical activity (“How could you evaluate your own physical activity?”) and perceived physical fitness (“How could you evaluate your own physical fitness when comparing with others?”) were assessed by the single questions developed in our previous study [34]. For perceived physical activity, the response options were from 0 (“I am not sufficiently active”) up to 3 (“I am very physically active”). For perceived physical fitness, the response options were from 0 (“I am very unfit”) up to 4 (“I am very fit”).

Physical activity was assessed by the Godin & Shepard Leisure-Time Exercise Questionnaire (LTEQ) [35]. The tool contains three questions with given examples of physical activities at strenuous (running, jogging, football, basketball), moderate (fast walking, easy bicycling, badminton, easy swimming), and light (yoga, archery, fishing from riverbank, golf, easy walking) intensity. Study participants were asked to indicate an approximate number of sessions of each intensity during the past week of a duration 15 min and more. If no sessions at the different intensities were performed during the past week, study participants had to indicate it as 0. The number of sessions at the strenuous, moderate, and light intensity was multiplied accordingly by 9, 5, and 3, and the results were summarized. The final score represented the higher physical activity at leisure-time.

Self-rated health was assessed by a single statement from the HBSC study [31]: “Would you say your health is” with the response options 1 “poor”, 2 “fair”, 3 “good”, and 4 “excellent”.

### 2.3. Statistical Analysis

After initial descriptive statistics and normality testing of continuous variables, the independent samples t-test was used to compare means between the two groups with the Cohen’s d to represent the effect sizes. The Hedge’s g correction was applied because of unequal sample sizes in boys and girls. Effect sizes above 0.2 were considered small, and equal or above 0.5 were considered moderate [36]. Next, Pearson or Spearman correlations were employed to test the associations between the study variables. The Pearson coefficient was used to test correlations between two continuous and normally distributed variables. For ordinal and not normally distributed continuous variables, the Spearman correlation coefficient was calculated. Magnitudes between 0.1 and 0.3 were considered small, above 0.3 and below 0.5 were considered moderate, and equal or above 0.5 were considered strong with a significance level of <0.05 [37]. All statistical analyses were carried out with SPSS v. 27 (IBM Corp., Armonk, NY, USA).

## 3. Results

The differences between the study measures in boys and girls are presented in Table 1. No significant differences were found only in self-objectification and weighting frequency. The mean body mass index, frequency of taking breakfast before school, and number of meals per day were higher in boys. On the other hand, girls demonstrated more frequent dieting and had a higher score of unhealthy eating habits such as eating in a rush, when reading or watching TV, late at night, and overeating. When comparing dietary habits, more frequent consumption of fruits, berries, and vegetables was observed in girls, whereas boys exhibited more frequent consumption of cereals and fish. In addition, the consumption of soft and energy drinks, and fast food was more frequent in boys, whereas girls more frequently consumed sweets and cakes. Next, the time of internet browsing was longer in girls, whereas sleep duration was longer in boys. In addition, boys demonstrated a higher leisure-time exercise score and greater perceived physical activity and physical fitness. Smoking and consuming alcohol until feeling dizzy were also more frequent in boys. All differences demonstrated small to medium effect sizes.

Next, we compared the mean self-objectification score in the groups of different meals for snacking between main eating (Table 2). In boys, no significant differences were found. In girls, in the group of the healthiest snacking (fruits, vegetables), the highest self-objectification score was detected.

Next, a series of correlations between self-objectification and lifestyle-related habits were conducted. To make it easier to interpret and summarize, we divided all lifestyles into three groups: eating attitudes and behaviors, dietary habits, physical activity-related and harmful behaviors. Thus, Table 3 represents the correlations between self-objectification and eating attitudes and behaviors. In girls, significant correlations were observed between self-objectification, body mass index, and dieting, whereas adverse correlations were found with breakfast frequency, no. of meals, and self-rated health. In contrast, in boys, a small but significant and positive correlation was detected between self-objectification and no. of meals per day. Importantly, significant negative correlations were observed between self-rated health, dieting, and unhealthy eating habits in boys and girls. In contrast, more frequent breakfasts and a higher number of meals per day correlated positively with self-rated health of adolescents.

Furthermore, Table 4 represents correlations between self-objectification and dietary habits. In boys and girls, self-objectification correlated with less heathy dietary habits, such as more frequent consumption of soft drinks, and energy drinks. In girls, an adverse correlation between self-objectification and consumption of cereals was found. In addition, positive correlation patterns were observed between health-beneficial dietary habits such as more frequent consumption of fruits, berries, vegetables, and cereals and between less health-beneficial dietary habits such as consuming soft and energy drinks and fast foods in boys and girls.

Finally, Table 5 provides correlations between self-objectification, harmful lifestyle habits, and physical activity-related time spending. Again, self-objectification correlated positively with smoking and alcohol consumption in boys and girls. Furthermore, in girls, a correlation between longer duration of internet browsing for non-educational purposes and self-objectification was found. In addition, girls with higher self-objectification demonstrated lower perceived physical activity. Importantly, medium correlations were found between perceived physical activity, fitness, and leisure-exercise in boys and girls. On the other hand, boys with greater fitness and leisure-exercise tended to use alcohol. In girls, longer sleep duration positively correlated with perceived physical activity and fitness, whereas in boys, there was a negative correlation with internet browsing duration for non-educational purposes.

## 4. Discussion

In the present study, we tested associations between self-objectification and health-related lifestyle habits in a large sample of adolescent girls and boys. Based on the objectification theory, conceptions of body functionality, and positive body image [1,5,6], we assumed that self-objectification would be positively associated with unhealthier eating and dieting, smoking, alcohol consumption and negatively associated with leisure-time exercise, perceived physical fitness, and self-rated health. We also expected the correlations would be higher in adolescent girls than in boys. The findings of the present study supported our hypotheses. In girls, higher self-objectification was associated with lower consumption of fruits and vegetables, lower breakfast frequency, eating frequency, higher dieting and weighing frequency, unhealthy eating habits, lower eating of cereals, and higher consumption of energy drinks. In boys, higher self-objectification was associated with higher eating frequency, drinking more soft drinks, energy drinks, and eating fast food.

The correlations of associations were low but significant, suggesting that adolescents appreciating body competence over appearance have more healthier eating habits, especially girls. Adolescent girls with lower self-objectification might have a more positive body image and express more love and care about their bodies [5]. Specifically, adolescent girls valuing body competence over appearance might be more likely to engage in healthy behaviours such as healthy eating. Another possible explanation of the results might be based on the acceptance model of intuitive eating. Intuitive eating is based on the self-care behaviour that involves listening to and trusting internal body cues such as hunger and safety to largely determine what, when, and how much to eat [38]. Intuitive eating is associated with a lower body mass index, less disordered eating, and healthier eating attitudes and behaviours [39,40]. Previously, it has been suggested that dieting is not consistent with internal psychological hunger and safety signals and disturbs interoceptive awareness, that is, the ability to identify, access, understand, and respond appropriately to the patterns of internal signals. Interoceptive awareness increases the susceptibility to other stimulus such as eating based on external or emotional motivation. Studies showed that a stronger emphasis on external factors such as appearance or body weight leads to a greater interference with energy intake regulation mechanisms [38]. We did not assess the intuitive eating in the present study; however, this study adds important new findings that self-objectification is associated with less favorable eating habits in adolescent girls. Future studies should continue to assess eating habits of adolescents considering the conceptions of body functionality and intuitive eating.

Furthermore, in the present study, we found that higher self-objectification was related to lower perceived physical activity in adolescent girls. A previous study also suggested that self-objectification prevents adolescent girls from performing physical activity [41]. According to the developmental theory of embodiment [42], joyful physical activity is one of the factors that helps promote positive embodiment and foster body functionality, helping women and girls resist sociocultural pressures to view their bodies as aesthetic objects. However, we did not find any associations between self-objectification and leisure–time exercise or perceived physical fitness. A previous review concluded that body image might be a correlate, antecedent, and the outcome of sport participation [43]. Sport participation might help decrease self-objectification for adolescent girls [2]; however, participation in so called weight-sensitive, especially aesthetic sports might put more emphasis on the appearance and increase self-objectification in adolescents [44]. However, these studies are lacking, and future studies addressing these associations in experimental or longitudinal designs are of great importance.

The present study adds important new findings suggesting that self-objectification is associated with tobacco smoking and alcohol consumption in adolescent girls and boys. Previous studies have found that negative body image is associated with higher alcohol consumption and smoking in adolescent girls but not in boys [45]. These results are in accordance with a longitudinal study showing that positive body image predicted lower smoking and alcohol consumption in adolescent girls [46]. Our study adds important new knowledge that self-objectification is associated with alcohol assumption and smoking in adolescent boys. Adolescent boys internalize muscular body image with low body fat, and previous studies suggested that drive for muscularity is associated with binge drinking [47].

The results of the present study showed that in girls, self-objectification was associated with longer hours of internet browsing. The detrimental effect of media consumption, especially of social media use, on the body image of adolescent girls is widely confirmed [48], and our results are in accordance of these findings. Stereotyped and unrealistic appearance standards are transmitted through media, and adolescents spending more time using media, especially social media, are more affected by these pressures; therefore, they value appearance over body competence. Girls spend more time using social networking compared to boys; therefore, they might be more affected by sociocultural pressures to attain body beauty ideals [49].

Finally, as predicted, the stronger associations between variables were observed in adolescent girls compared to boys, suggesting that self-objectification is more detrimental for the lifestyle of adolescent girls than boys. These findings are consistent with the main tenets of objectification theory [1,2]. An important new finding of the present study is that self-objectification in adolescent girls was associated with lower self-rated health. Self-rated health is an important indicator of physical and psychological well-being [50], and our study adds important new evidence that valuing appearance over body competence might be an important factor in decreasing it.

The findings of the present study reflected general tendencies in the lifestyle of adolescent girls and boys [51,52]. Adolescent boys reported eating breakfast more frequently compared to girls; they also ate more frequently, reported less dieting, used less unhealthy dietary habits, ate less fruits and vegetables but ate more cereals and fish, used higher amounts of soft and energy drinks, ate more fast food, used more alcohol until feeling dizzy, smoked more, and browsed internet longer compared to girls. Nevertheless, adolescent boys were more physically active, reported higher perceived physical fitness, and reported higher self-rated health. However, in the present study, we did not observe significant differences in adolescent self-objectification. It contradicts previous findings [2].

The present study has important limitations that should be discussed. First, the self-objectification in the present sample was low; therefore, it might affect the results. Generalization of our findings should also be limited because the sample was from Lithuania. Future studies should test these associations in samples of adolescents in other countries. Furthermore, the present study is cross-sectional and associations between variables are bidirectional. However, findings of the previous studies suggested that the dysfunctional eating and negative body image is an outcome but not a precedent of self-objectification [7,20]. Nevertheless, future studies of longitudinal designs should test our findings. Finally, self-objectification (valuing appearance over body competence) was tested using a self-objectification questionnaire [20] that measures appearance and body competence-related body attributes. Future studies might benefit from assessing body functionality that consists of broad domains such as creative endeavors, bodily senses and sensations, communication with others, and self-care [6].

The results of the present study inform healthy lifestyle promotion programs for adolescents, suggesting that including psychoeducation about self-objectification, body functionality, and positive embodiment might be an effective strategy. In other words, promoting healthy eating and physical activity and preventing alcohol consumption and smoking might be more effective if they are implemented together with teachings about positive body image, body functionality, and resistance towards sociocultural pressures.

## 5. Conclusions

This cross-sectional study in a large adolescent sample of both genders adds important knowledge that self-objectification is associated with a poorer health-related lifestyle. In girls, higher self-objectification was associated with lower consumption of fruits and vegetables, lower breakfast and eating frequency, higher dieting and weighing frequency, unhealthy eating habits, lower eating of cereals, higher consumption of energy drinks, lower perceived physical activity, longer hours of internet browsing, and lower self-rated health. In boys, higher self-objectification was associated with higher eating frequency, drinking more soft drinks, energy drinks, and eating fast food. The present study adds important new findings that self-objectification is associated with tobacco smoking and alcohol consumption in adolescent girls and boys. The results of the present study inform healthy lifestyle promotion programs for adolescents, suggesting that including psychoeducation about self-objectification, body functionality, and positive embodiment might be an effective strategy.

## Figures and Tables

**Table 1 children-09-01022-t001:** The comparison of the self-objectification and lifestyle habits in adolescent boys and girls (*n* = 1412).

Characteristics	Range	Boys (*n* = 570)		Girls (*n* = 1412)		*t, p*	Cohen’s d
		m	SD	m	SD		
Self-objectification	−25–25	−6.62	9.55	−5.95	10.78	−1.22, 0.22	-
Body mass index	14.0–41.7	21.99	2.99	20.96	3.06	6.27, <0.001	0.34
Breakfast frequency	0–3	2.04	1.06	1.88	1.14	2.56, 0.01	0.14
No. of meals/day	1–10	3.79	1.21	3.29	1.13	8.06, <0.001	0.43
Dieting	0–4	0.42	0.86	0.90	1.12	−9.12, <0.001	0.47
Weighting frequency	0–6	1.62	1.52	1.57	1.52	0.61, 0.542	-
Unhealthy dietary habits	0–4	1.69	0.66	1.82	0.61	−3.71, <0.001	0.21
Self-rated health	1–4	3.02	0.81	2.72	0.78	6.75, <0.001	0.38
Fruits, berries	0–3	1.59	0.83	1.70	0.88	−2.33, 0.02	0.13
Vegetables	0–3	1.68	0.82	1.81	0.86	−0.92, 0.004	0.15
Cereals	0–3	1.85	0.84	1.73	0.88	2.54, 0.01	0.14
Fish	0–3	0.69	0.70	0.53	0.66	4.33, <0.001	0.24
Sweets, cakes	0–3	1.33	0.81	1.66	0.85	−7.29, <0.001	0.40
Soft drinks	0–3	1.04	0.81	0.75	0.81	6.38 <0.001	0.36
Energy drinks	0–3	0.53	0.76	0.30	0.62	6.00, <0.001	0.34
Fast food	0–3	0.92	0.74	0.72	0.71	5.19, <0.001	0.28
Sleep duration, hours	3–14	7.43	1.38	7.07	1.28	5.04, <0.001	0.27
Smoking	0–3	0.79	1.17	0.57	1.03	3.67, <0.001	0.20
Alcohol until feeling drunk	0–4	1.77	1.54	1.53	1.48	3.00, 0.003	0.16
Browsing internet, hours/day	0–10	3.80	2.31	4.30	2.29	−4.08, <0.001	0.22
Perceived physical activity	0–3	1.77	1.02	1.14	1.02	11.38, <0.001	0.62
Perceived physical fitness	0–4	2.32	1.03	1.95	0.98	6.71, <0.001	0.37
Leisure-time exercise score	0–200	77.34	45.57	61.31	42.04	6.69, <0.001	0.37

m = mean, SD = standard deviation.

**Table 2 children-09-01022-t002:** The comparison of self-objectification in the groups of snacking habits in adolescent boys and girls (*n* = 1412).

Food for Snacking	Boys (*n* = 570)		Girls (*n* = 1412)	
	m	SD	m	SD
Fruits and vegetables	−7.24	9.54	−7.16	11.25
Sweets, cakes, chips	−6.29	9.68	−5.48	10.59
No snacking	−6.65	8.45	−2.33	10.01
F, *p*	0.59; 0.55		4.27; 0.014	

m = mean, SD = standard deviation.

**Table 3 children-09-01022-t003:** Associations between self-objectification, body mass index, self-rated health, and eating-related attitudes and behaviors in adolescent boys and girls (*n* = 1412).

Correlates (Girls, *n* = 842)	SO	BMI	BR	ML	DIE	WEI	UEH	SRH
Self-objectification (SO)	1							
Body mass index (BMI)	0.15 **	1						
Breakfast frequency (BR)	−0.16 **	−0.13 **	1					
No. of meals/day (ML)	−0.10 **	−0.09 *	0.40 **	1				
Dieting (DIE)	0.27 **	0.30 **	−0.20 **	−0.17 **	1			
Weighting frequency (WEI)	0.11 **	0.06	−0.07 *	−0.04	0.22 **	1		
Unhealthy dietary habits (UEH)	0.09 **	0.01	−0.09 **	0.02	0.002	−0.05	1	
Self-rated health (SRH)	−0.11 **	−0.04	0.15 **	0.16 **	−0.18 **	−0.03	0.15 **	1
Correlates (boys, *n* = 570)								
Self-objectification (SO)	1							
Body mass index (BMI)	0.01	1						
Breakfast frequency (BR)	−0.01	0.04	1					
No. of meals/day (ML)	0.14 **	0.04	0.22 **	1				
Dieting (DIE)	0.01	0.31 **	−0.09 *	0.01	1			
Weighting frequency (WEI)	0.04	0.12 *	0.04	0.08	0.24 **	1		
Unhealthy eating habits (UEH)	0.05	−0.01	−0.12 *	−0.09 *	−0.02	−0.05	1	
Self-rated health (SRH)	−0.01	−0.03	0.12 **	0.11 *	−0.09 *	0.10 *	−0.14 **	1

* *p* < 0.05, ** *p* < 0.01.

**Table 4 children-09-01022-t004:** Associations between self-objectification and dietary habits in adolescent boys and girls (*n* = 1412).

Correlates (Girls, *n* = 842)	SO	FB	VG	CR	FS	SW	SD	ED	FF
Self-objectification (SO)	1								
Fruits, berries (FB)	−0.06	1							
Vegetables (VG)	−0.04	0.48 **	1						
Cereals (CR)	−0.12 **	0.22 **	0.23 **	1					
Fish (FS)	−0.01	0.17 **	0.15 **	0.05	1				
Sweets, cakes (SW)	0.01	−0.05	0.02	0.18 **	−0.02	1			
Soft drinks (SD)	0.09 *	−0.15 **	−0.11 **	−0.03	0.003	0.24 **	1		
Energy drinks (ED)	0.15 **	−0.04	−0.07 *	−0.13 **	0.06	−0.02	0.25 **	1	
Fast food (FF)	0.04	−0.16 **	−0.09 **	−0.01	0.001	0.21 **	0.43 **	0.22 **	1
Correlates (boys, *n* = 570)									
Self-objectification (SO)	1								
Fruits, berries (FB)	−0.08	1							
Vegetables (VG)	−0.05	0.44 **	1						
Cereals (CR)	−0.05	0.22 **	0.31**	1					
Fish (FS)	0.09 *	0.18 **	0.15**	−0.03	1				
Sweets, cakes (SW)	0.07	0.10 *	0.09*	0.14 **	0.04	1			
Soft drinks (SD)	0.13 **	−0.08 *	−0.07	−0.03	0.01	0.33 **	1		
Energy drinks (ED)	0.09 *	−0.11 **	−0.18**	−0.21 **	0.16 **	0.07	0.39 **	1	
Fast food (FF)	0.10 *	−0.06	−0.08*	0.01	0.08 *	0.22 **	0.43 **	0.38 **	1

* *p* < 0.05, ** *p* < 0.01.

**Table 5 children-09-01022-t005:** Associations between self-objectification, harmful lifestyle habits, sleep duration, and physical activity-related behaviors in adolescent boys and girls (*n* = 1412).

Correlates (Girls, *n* = 842)	SO	SL	SM	ALC	BR	PPA	PPF	LE
Self-objectification (SO)	1							
Sleep duration (SL)	−0.05	1						
Smoking (SM)	0.17 **	−0.03	1					
Alcohol until feeling drunk (ALC)	0.17 **	−0.05	0.58 **	1				
Internet browsing duration (BR)	0.09 *	−0.05	0.17 *	0.17 *	1			
Perceived physical activity (PPA)	−0.11 *	0.19 **	−0.01	−0.001	−0.09 *	1		
Perceived physical fitness (PPF)	−0.05	0.13 **	−0.01	0.02	−0.06	0.46 **	1	
Leisure-time exercise (LE)	−0.01	0.05	0.03	0.05	0.03	0.36 **	0.25 **	1
Correlates (boys, *n* = 570)								
Self-objectification (SO)	1							
Sleep duration (SL)	−0.001	1						
Smoking (SM)	0.08 *	−0.07	1					
Alcohol until feeling drunk (ALC)	0.11 *	−0.01	0.53 **	1				
Internet browsing duration (BR)	0.00	−0.14 **	0.08	0.16 **	1			
Perceived physical activity (PPA)	0.01	0.03	−0.04	0.02	−0.14 **	1		
Perceived physical fitness (PPF)	0.04	0.05	0.03	0.11 **	−0.09 *	0.57 **	1	
Leisure-time exercise (LE)	−0.02	0.02	0.06	0.12 **	−0.02	0.38 **	0.26 **	1

* *p* < 0.05, ** *p* < 0.01.

## Data Availability

The dataset generated and analyzed during the current study is not publicly available but is available from the corresponding author on reasonable request.

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
