# Peer review of "Associations between Self-Objectification and Lifestyle Habits in a Large Sample of Adolescents"

_children, 2022, doi:10.3390/children9071022_

Round 1

Reviewer 1 Report

Dear Authors,

your article "Associations between Self-Objectification and lifestyle habits in a large sample of adolescents" addresses a topic of great importance, especially today, when adolescents are more vulnerable due to the challenges generated by the social digital transformations. The research you've made is complex and I appreciate a lot the fact that it captures from multiple perspectives the current problems of adolescents. I have no comments about the way you collected and analysed the data, generally regarding the methodology you used. Also, the explanations are clear and well sustained by the data. The only comment I could make regards "the aim" of the study (respectively the lines 63 to 69) which is explained too succinctly  compared to the detailed explanations of each of the dimensions that associate with the issues of self-objectification in adolescence. 

So, I consider this study could be published as it is, or after minor reviewing of the way the hypotheses are explained through the above mentioned paragraph. 

Hoping that the results of your study will be used by third-party stakeholders and educational institutions, and also that other researcher will deepen these subjects, I congratulate you for the initiative and this article.

Best regards!   

Author Response

Dear Reviewer,

Thank you for your time reviewing our paper and for your comments. All changes made in the text are highlighted in a blue font.

Dear Authors,

your article "Associations between Self-Objectification and lifestyle habits in a large sample of adolescents" addresses a topic of great importance, especially today, when adolescents are more vulnerable due to the challenges generated by the social digital transformations. The research you've made is complex and I appreciate a lot the fact that it captures from multiple perspectives the current problems of adolescents. I have no comments about the way you collected and analysed the data, generally regarding the methodology you used. Also, the explanations are clear and well sustained by the data.

Thank you for this comment.

The only comment I could make regards "the aim" of the study (respectively the lines 63 to 69) which is explained too succinctly compared to the detailed explanations of each of the dimensions that associate with the issues of self-objectification in adolescence. 

This comment is not clear for us. We assume that the aim of the study should be maximally clear and short.

So, I consider this study could be published as it is, or after minor reviewing of the way the hypotheses are explained through the above mentioned paragraph. 

Hoping that the results of your study will be used by third-party stakeholders and educational institutions, and also that other researcher will deepen these subjects, I congratulate you for the initiative and this article.

Best regards!   

Thank you once again for warm words!

Reviewer 2 Report

Dear Authors,

The manuscript entitled “Associations between Self-Objectification and Lifestyle Habits in a Large Sample of Adolescents” deals with an interesting and important issue that perfectly fits in the journal’s scope. However, you should consider the following comments to improve it.

The Introduction section needs to be expanded with relevant literature sources; since the number of studies on adolescents is relatively low, I recommend you to include research results on young adults (e.g., higher education students), but even results of research on adults could be interesting and useful.

Regarding the sample, what can we know about its representativeness? What was the response rate? How could you control who answered the questions and that one respondent answered only once?

The Self-Objectification Questionnaire contains 2*5 questions, each ranging from 0 to 9, therefore, the maximum of the two subscales is 5*9=45 each, and the minimum is 0*9=9 each. Thus, the theoretical maximum of the difference of the two subscales is 45-0=45, the theoretical minimum is 0-45=-45. So the final score range was not from -25 to +25. (Maybe this range is the range of the actual scores, but in this case, please, make it clear.)

In what time period should participants provide their sleep duration and internet browsing duration? In the last one week? Or in the last one month? Or in the last half year? It is necessary to specify the time period in which a daily average is asked.

It is unnecessary to explain the meaning of self-objectification all the time the term appears in the text; it is enough to explain it for the first time, and mention its meaning if it is important.

There are only a few wording issues that need to be addressed: in the abstract, line 15 needs a revision; the term “gymnasium” is not the best choice, it is better to use high school or secondary school. What is the difference between soft drinks and soda? In my understanding, soda is a type of soft drinks, therefore the use of these two unhealthy food groups together is misleading.

Author Response

Dear Reviewer,

Thank you for your time reviewing our paper and for your valuable comments. All changes made in the text are highlighted in a blue font.

Dear Authors,

The manuscript entitled “Associations between Self-Objectification and Lifestyle Habits in a Large Sample of Adolescents” deals with an interesting and important issue that perfectly fits in the journal’s scope. However, you should consider the following comments to improve it.

Thank you for this comment.

The Introduction section needs to be expanded with relevant literature sources; since the number of studies on adolescents is relatively low, I recommend you to include research results on young adults (e.g., higher education students), but even results of research on adults could be interesting and useful.

Thank you for this comment. We have included the main findings in adults to the Introduction.

Regarding the sample, what can we know about its representativeness? What was the response rate? How could you control who answered the questions and that one respondent answered only once?

Thank you for this comment. We cannot assert that the sample is perfectly country-representative as the recruitment of the schools was not probabilistic. However, 41 school from 26 cities and towns covered all main country municipalities.

The response rate was close to 90%. As the survey was anonymous, there was no possibility to control who had provided their forms. The online survey form was restricted to accept only one answer from the same IP address. We have added some more information to the section 2.1.

The Self-Objectification Questionnaire contains 2*5 questions, each ranging from 0 to 9, therefore, the maximum of the two subscales is 5*9=45 each, and the minimum is 0*9=9 each. Thus, the theoretical maximum of the difference of the two subscales is 45-0=45, the theoretical minimum is 0-45=-45. So the final score range was not from -25 to +25. (Maybe this range is the range of the actual scores, but in this case, please, make it clear.)

Thank you for this comment. In this scale, each body feature can be given a score only once. If someone gives a score of nine to “health”, it means that nine cannot be given once again to another body feature. Below, the first example provides scoring in a case of the lowest possible self-objectification (SO) when a person ranks all appearance-related body features by the lowest values. Thus, after summarizing the scores for appearance (APP) and competence (COMP) domains and subtracting competence attributes from appearance, we get -25, which means the lowest possible SO. The second scoring example represents an opposite scenario when a responder ranks in the highest scores in all body appearance features and collects the maximum score of +25, which is described as the highest possible SO. Of course, most of the respondents in our sample scored higher in competence-related body features as the sample means of the SO is negative.

APP               COMP

0 2 4 3 1      5 7 8 9 6

10-35 = -25 (the lowest possible self-objectification);

9 6 7 8 5      1 0 3 4 2

35-10 = 25 (the highest possible self-objectification).

We have added an additional clarification to the scale description: “The same score could be given only once. Thus, each body feature had to be attributed a different score.”

In what time period should participants provide their sleep duration and internet browsing duration? In the last one week? Or in the last one month? Or in the last half year? It is necessary to specify the time period in which a daily average is asked.

Thank you for this comment. We have added information to the Methods description: “Sleep duration in hours per usual workday was indicated by the study participants” and “The duration of internet browsing for non-educational purposes was assessed by a single question asking to indicate the mean time in hours per usual workday”.

It is unnecessary to explain the meaning of self-objectification all the time the term appears in the text; it is enough to explain it for the first time, and mention its meaning if it is important.

Thank you for this comment, we double checked all text and removed redundant information.

There are only a few wording issues that need to be addressed: in the abstract, line 15 needs a revision;

Thank you for this comment, the sentence was revised.

 the term “gymnasium” is not the best choice, it is better to use high school or secondary school.

Thank you for this comment, we change “gymnasium” into “school”.

What is the difference between soft drinks and soda? In my understanding, soda is a type of soft drinks, therefore the use of these two unhealthy food groups together is misleading.

Thank you for this comment. We changed this unhealthy food group title into “soft drinks”.

Sincerely,

The authors

Reviewer 3 Report

Accept after minor revision

I believe the manuscript is fascinating since it tested associations between self-objectification and health-related lifestyle habits in adolescents. It provides support for the healthy growth of adolescents. The comments are as follows:

1. Line 45: “The self-objectification theory is less tested on adolescents;…” Can the authors add to the current state of research on SO in other populations?

2. Lines 174-175: “Next, Pearson or Spearman correlations were employed to test the associations between the study variables. ” What situations are analyzed using Person?

3. Line 365: It is recommended that the author provide an overview of the content and methods of manuscript research in conclusions.

Author Response

Dear Reviewer,

Thank you for your time reviewing our paper and for your comments. All changes made in the text are highlighted in a blue font.

I believe the manuscript is fascinating since it tested associations between self-objectification and health-related lifestyle habits in adolescents. It provides support for the healthy growth of adolescents.

Thank you for this comment.

The comments are as follows:

  1. Line 45: “The self-objectification theory is less tested on adolescents;…” Can the authors add to the current state of research on SO in other populations?

Thank you for this comment. We have included the main findings in adults to the Introduction.

  1. Lines 174-175: “Next, Pearson or Spearman correlations were employed to test the associations between the study variables. ” What situations are analyzed using Person?

Thank you for this comment. We added an additional clarification to the Statistical analysis section: “Pearson coefficient was used to test correlations between two continuous and normally distributed variables. For ordinal and not normally distributed continuous variables Spearman correlation coefficient was calculated”.

  1. Line 365: It is recommended that the author provide an overview of the content and methods of manuscript research in conclusions.

Thank you for this comment. We revised the first sentence of the conclusions addressing this comment.

Sincerely,

The authors

Round 2

Reviewer 2 Report

Dear Authors,

Thank you for the responses to my questions. All of my concerns have been addressed; however, I would read a little bit more about previous research results in the Introduction section. Besides, only minor grammatical issues remained in the manuscript that need to be checked before publication.